# A Systematic Review of Vitamin D Supplementation in Oncology: Chance of Science or Effectiveness?

**DOI:** 10.3390/nu17040634

**Published:** 2025-02-11

**Authors:** Marta Lourenço Afonso, Manuel Luís Capelas, Nuno M. Pimenta, Teresa Santos, Antti Mäkitie, Susana Ganhão-Arranhado, Carolina Trabulo, David da Silva Dias, Pedro Miguel Neves, Paula Ravasco

**Affiliations:** 1Universidade Católica Portuguesa, Faculty of Health Sciences and Nursing, 1649-023 Lisbon, Portugal; luis.capelas@ucp.pt; 2Universidade Católica Portuguesa, Faculty of Health Sciences and Nursing, Centre for Interdisciplinary Research in Health (CIIS), 1649-023 Lisbon, Portugal; npimenta@esdrm.ipsantarem.pt (N.M.P.); tsantos@ucp.pt (T.S.); antti.makitie@helsinki.fi (A.M.); sarranhado@uatlantica.pt (S.G.-A.); carolinafptrabulo@gmail.com (C.T.); daviddias_77@hotmail.com (D.d.S.D.); pedrompneves@gmail.com (P.M.N.); pravasco@ucp.pt (P.R.); 3Polytechnic Institute of Santarém, Sport Sciences School of Rio Maior, 2040-413 Rio Maior, Portugal; 4Interdisciplinary Centre for the Study of Human Performance, Faculdade de Motricidade Humana, Universidade de Lisboa, 1499-002 Lisbon, Portugal; 5Universidade Europeia, Psychology Department, 1500-210 Lisboa, Portugal; 6Department of Otorhinolaryngology-Head and Neck Surgery, Helsinki University Hospital, University of Helsinki, 00029 HUS Helsinki, Finland; 7Research Program in Systems Oncology, Faculty of Medicine, University of Helsinki, 00014 Helsinki, Finland; 8Division of Ear, Nose and Throat Diseases, Department of Clinical Sciences, Intervention and Technology, Karolinska Institute, Karolinska University Hospital, 171 76 Stockholm, Sweden; 9Atlântica, Instituto Universitário, Fábrica da Pólvora de Barcarena, 2730-036 Barcarena, Portugal; 10CINTESIS, Centre for Health Technology and Services Research, 4200-450 Porto, Portugal; 11Medical Oncology, Centro Hospitalar Barreiro-Montijo, 2830-003 Barreiro, Portugal; 12Medical Oncology Department, Unidade Local de Saúde Cova da Beira, 6200-251 Covilhã, Portugal; 13Faculty of Health Sciences, Universidade da Beira Interior, 6200-251 Covilhã, Portugal; 14Universidade Católica Portuguesa, Faculty of Medicine, 2635-631 Rio de Mouro, Portugal; 15Centre for Interdisciplinary Research in Health Egas Moniz (CiiEM), 2829-511 Almada, Portugal

**Keywords:** adverse effects, cancer, neoadjuvant therapy, neoplasm, vitamin D

## Abstract

Background: Vitamin D (VD) supplementation has increased considerably in the last decade, whether for the prevention or treatment of numerous diseases, including bone, cardiovascular, endocrine, neurologic, psychological, respiratory, infectious, or oncological. The primary objective of this scoping review was to examine and synthesize the scientific evidence on the role of VD in all-type cancer patients undergoing adjuvant and neoadjuvant therapy with chemotherapy (CT) or radiotherapy (RT), namely in improving side effects. Methods: This review was conducted by selecting papers from the CINAHL, Scopus and PubMed databases based on the descriptor terms mesh and title/abstract, taking into consideration the defined inclusion and exclusion criteria, following the PRISMA-ScR (PRISMA extension for scoping reviews) statement. Results: A total of 758 papers were identified in different databases during this review. However, using the inclusion and exclusion criteria, only five publications made up the final sample of the study. The studies included heterogeneous study methodologies, objectives, cancer diagnosis, as well as methods to assess body composition, which makes it difficult to compare them. Based on the analyzed studies, associations were found between bone density and VD in patients who underwent preoperative chemoradiotherapy (CRT). In patients with non-small-cell lung cancer receiving CT, some of the side effects associated with the treatment were attenuated and reduced. In addition, another of the studies analyzed found that VD deficiency (VDD) has been associated with increased peripheral neuropathy (PN) induced by CT in the treatment of breast cancer. VD supplementation was found to be safe and effective. Conclusions: In this scoping review, VD is highlighted as a crucial factor in preventing the side effects of neoadjuvant RT or CT, as well as treating other treatment-related health conditions, such as osteoporosis, as well as ameliorating the side effects (nausea, vomiting, fatigue) associated with aggressive CT and RT.

## 1. Introduction

Vitamin D (VD) is a fat-soluble vitamin that acts as a precursor to 1,25-dihydroxyvitamin D in the body [1,2,3,4,5,6,7,8,9,10]. Its main functions include modulating calcium [9,11,12,13] and phosphorus homeostasis and osteosynthesis [11,12,13], bone structure [12,14,15,16], and physiological functions, as well as supporting nerve, immune, and muscle health [10,17,18]. The minimum level of 25-hydroxyvitamin D (25(OH)D) in serum is 30 ng/mL (75 nmol/L) [3,7,8,10]. Below this value, patients suffer from mild (29–20 ng/mL), moderate (19–10 ng/mL), or severe VD deficiency (VDD) [4,10]. Thus, the guidelines recommend daily supplementation with 1200 mg of calcium and 400 IU of VD [4,7]. However, this dose of VD is not sufficient to normalize the 25-dihydroxyvitamin D level in most patients undergoing adjuvant chemotherapy (CT) [7,8]. Over the past decade, VD supplements have been used to prevent or treat bone, cardiovascular, endocrine, neurological, psychiatric, respiratory, and infectious diseases or cancer [1,2].

Several studies have suggested that VDD increases the risk of infections, diabetes mellitus types 1 and 2, cardiovascular diseases, obesity, asthma, inflammatory bowel disease, colon, breast, prostate, and ovarian cancer, as well as some neurological conditions [1,6,19]. Apoptosis, invasion, stimulation of cell differentiation, inhibition of angiogenesis, anti-inflammatory and anti-proliferative effects, and metastasis are some ways VD influences the natural history of cancer [4,10,11]. Researchers have found that VDD is common among cancer patients [7,19]. In these patients, the prevalence of VDD varies from 14% to 92% [19]. CT and radiotherapy (RT) treatment may result in VDD as well as changes to bone metabolism, leading to osteoporosis and other skeletal problems, such as osteomalacia, increased fracture risk, decreased bone mineral density, avascular necrosis, and skeletal deformities [19,20,21,22]. Neoadjuvant therapy is the preoperative induction treatment of tumors with CT, RT, and endocrine therapy [22,23]. The correction of VDD is important [24] since low VD levels are associated with cancer prognosis in patients receiving adjuvant CT or neo-adjuvant RT [7,24]. It has been shown that VDD is strongly associated with cancer risk. Thus, VD supplementation may be crucial to cancer patients’ lives. By preventing VDD, alleviating cancer-related adverse symptoms, such as fatigue, vomiting, and nausea, and improving cancer incidence, prognosis, and outcomes, VD supplementation can help prevent VDD from occurring [7,24].

Most VD supplements and fortified foods do not pose a risk of toxicity unless taken in very large doses. Physiological mechanisms limit the formation and metabolism of the vitamin [11,14,25,26,27]. Its original purpose was to reduce tumors and allow a mastectomy in place of a breast-conserving procedure. Cancer treatment is increasingly relying on it [22,23]. Originally defined as systemic therapy given before local treatment, the concept has been extended to include radiotherapy given before surgery [28]. Advantages may include better local and remote control, direct assessment, and organ preservation treatment [22]. On the other hand, potential disadvantages are the increased toxicity and cost, potential delay in effective treatment, and obscuring pathological staging [28,29,30].

By interfering with healthy cells in the body, CT can lead to the appearance of unwanted symptoms [22,29,31]. Adverse effects of CT can last a few days to a few weeks [28,32] and include hormonal and mood changes, anemia, absence or decrease in appetite, gastrointestinal adverse effects (gastrointestinal toxicity, vomiting, nausea, constipation, dysphagia, aversion to the smell or taste of food, anorexia), fatigue, mouth sores and opportunistic infections, neurotoxicity, and hematological toxicity [8,28,30,33]. Long-term CT can also cause changes in the heart, liver, lungs, and nervous system. It is pertinent to note that patients will experience different effects depending on the drug used. A treatment cycle’s effects are not necessarily repeated in subsequent cycles [24,31]. A new strategy in neoadjuvant therapy, like other innovations in cancer treatment, must be evaluated closely against standard therapy for recurrence, survival, and impact on organ economy, as well as for quality of life and costs [24]. An adjuvant approach is usually recommended when a patient with early-stage cancer undergoes surgery or radiotherapy if this is considered to improve the prognosis and decrease the risk of recurrence. It is often determined based on the type and stage of the cancer whether the patient is a candidate for further treatment [24].

This study aimed to conduct a scoping review, which is a type of evidence synthesis that systematically identifies and maps the breadth of evidence available on a particular topic within or across particular contexts [34]. A scoping review can clarify key concepts in the literature and identify key related aspects, including those related to methodological research [34]. As with systematic reviews, this scoping review, of an exploratory nature, followed the PRISMA extension for scoping reviews [35,36] aiming to examine and synthesize the scientific evidence on the role of VD in cancer patients with any type of cancer undergoing adjuvant and neoadjuvant therapy with CT or RT, namely in improving side effects.

Thus, the guiding question for this paper was, “What is the role of Vitamin D in the side effects of adjuvant and neoadjuvant radiotherapy and chemotherapy in cancer patients?”.

## 2. Materials and Methods

### 2.1. Search Strategy

The search was conducted through MEDLINE (via PubMed, National Library of Medicine, Bethesda, Maryland), CINAHL (via EBSCO, Ipswich, MA, USA), and Scopus (via Elsevier, Amsterdam, The Netherlands) databases, according to the descriptor terms mesh and title/summary described in the databases until 20 August 2022 (Appendix A).

As a starting point for this scoping review, the guiding question was formulated following the steps of the methodology defined by The Joanna Briggs Institute Reviewers’ Manual 2015 formulation of the research question, according to the PCC model, which includes Population (P), Concept (C), and Context (C). In this review, the population focused on adult cancer patients, the concept addressed the adverse effects of vitamin D supplementation, and the context examined its effects on adjuvant and neoadjuvant radiotherapy and chemotherapy. The literature search was oriented to answer the following advanced question: what is the role of Vitamin D in the side effects of adjuvant and neoadjuvant RT and CT in cancer patients [35]?

The search in databases was conducted using Boolean operators (OR and AND), parentheses, quotation marks, and asterisks. Quotation marks were used to search for exact terms or expressions; parentheses were used to indicate a group of search terms or combine two or more groups of search terms, enabling all possible combinations of sentences; asterisks (*) were used to search all words derived from the precedent inflected part. Any filters to refine the search were not added. The searches were conducted in the online database, and the results were exported to a Microsoft Office Excel^®^ 2016 document (Microsoft Corporation, Redmond, WA, USA).

### 2.2. Selection Criteria

Titles and abstracts of all potentially relevant papers, and subsequently, full text, were reviewed to choose those that were eligible. Two investigators conducted both steps independently, and discordant results were resolved through consensus. The eligibility and exclusion criteria were defined to direct the search to answer the established question.

As inclusion criteria, we included studies published in the last five years that included adults (≥18 years old) with a cancer diagnosis submitted to adjuvant and neoadjuvant RT or CT and that mentioned one of the side effects, such as hormonal and mood changes, anemia, absence or decrease in appetite, gastrointestinal adverse effects (gastrointestinal toxicity, vomiting, nausea, constipation, dysphagia, aversion to the smell or taste of food, anorexia), fatigue, mouth sores and opportunistic infections, neurotoxicity, and hematological toxicity.

Focusing on recent studies ensures that the data reflect the most up-to-date state of scientific knowledge. Including studies from longer periods could excessively broaden the scope of the review, making the analysis and synthesis of results more challenging. Restricting the inclusion to the last five years allows for a more targeted and manageable focus, which is particularly important in a systematic review.

This approach is especially relevant given the probable increase in interest in vitamin D supplementation during this period, likely driven by the COVID-19 pandemic. The pandemic highlighted the role of vitamin D in immune modulation and disease outcomes, leading to a surge in research into its applications, including oncology.

As exclusion criteria, we excluded articles with patients with hematologic tumors because they have significantly different biological and pathophysiological characteristics than solid tumors. This can result in very different responses to VD supplementation, introducing high variability in the results and making comparative analysis between studies difficult. In addition, we also excluded studies that did not focus on the relationship between VD (interventions in which VD is not used alone) and the treatment of patients with cancer because interventions that combine VD with other therapeutic factors make it difficult to determine the isolated impact of VD on clinical outcomes. This confounding of variables can compromise the validity of conclusions about the specific role of vitamin D.

The duplicates were excluded, and titles and abstracts of all potentially relevant papers, and subsequently, full text, were reviewed to choose those that were eligible.

These criteria help to ensure greater homogeneity, robustness, and relevance of the results for the target population studied.

The scoping review followed the normative steps of the same enunciated in the PRIS-MA-ScR statement (Preferred Reporting Items for Systematic Reviews and Meta-Analysis extension for scoping reviews) [36].

### 2.3. Data Extraction

The data extraction of each study was based on the following information: the age of the population, the years examined, inclusion and exclusion criteria, cancer treatments, and VD supplementation.

## 3. Results

The flow diagram of included studies is represented in Figure 1. In the search of the MEDLINE, CINAHL, and Scopus databases, 758 records were identified. However, no papers were found in the CINAHL database after applying the search terms. After applying the inclusion and exclusion criteria, 643 papers were excluded. Thus, 115 papers were considered for eligibility, and 94 of these papers were excluded. The reading of the papers’ titles and abstracts allowed for the selection of 21 publications of potential interest, which were fully analyzed. After this review, the final sample consisted of five studies (Appendix B).

Yamada et al. observed that after dividing patients into two groups based on changes in bone density, the group with low bone density exhibited a shorter metastasis-free survival time (DMFS) after surgery compared with the other group (*p* < 0.05) [37]. Univariate analysis identified decreased VD concentration (<20 pg/mL) as a risk factor for changes in bone density (*p* = 0.04) [37]. Furthermore, multivariate analysis revealed that preoperative CRT was the only factor significantly associated with altered bone density (±, OR: 5.8; *p* = 0.04) [37]. These findings highlight a potential link between VD levels and bone health in oncological patients, warranting further investigation.

Similarly, Laviano et al. examined the adverse event profiles of two groups, one receiving juice-based TMN (target medical nutrition) and the other an isocaloric comparator. The TMN group (mean age 65 years) experienced fewer adverse events, including diarrhea, nausea, vomiting, and neutropenia, compared with the comparison group (64 vs. 87 events, respectively) [38]. Although compliance was slightly lower in the TMN group (58.5%) compared with the comparator group (73.6%), safety parameters such as blood pressure and heart rate favored the TMN intervention. These findings suggest that TMN, which included VD, may have protective effects on vital signs and overall tolerance of treatment [38].

In a broader analysis, Alkan et al. investigated VDD prevalence in a cohort of 706 cancer patients, revealing that 72% (509 patients) were vitamin D deficient. Multivariate analysis identified several factors associated with VDD, including low sun exposure (OR: 1.4 [95% CI: 1.009–2.1], *p* = 0.045), palliative care status (OR: 1.5 [95% CI: 1.008–2.4], *p* = 0.04), and history of gastrointestinal surgery [6]. Notably, adjuvant treatment initially appeared as a risk factor (OR: 2.6 [95% CI: 1.3–5.1], *p* = 0.006); however, the small *p*-value indicated that the result might not be robust, leading to its rejection in this study. These data underscore the multifactorial nature of VDD in cancer patients and the need for targeted interventions.

Further emphasizing the implications of VD levels, Jennaro et al. reported that VD-deficient patients exhibited a higher mean increase in peripheral neuropathy (PN) than non-deficient patients, along with a non-significant trend toward increased risk of treatment discontinuation [19]. Multivariate models revealed an inverse association between baseline VD levels and PN, highlighting the potential protective role of adequate VD levels in mitigating this common treatment-related side effect.

Conversely, Chartron et al. focused on the effectiveness and safety of VD supplementation in breast cancer (BC) patients undergoing adjuvant CT. In the D1C6 phase of their study, 21 patients achieved normalization of VD levels, with high-dose VD supplementation increasing the normalization rate of serum 25-OH VD to 47%. VD and calcium supplementation were well tolerated, with no clinical toxicity events related to treatment. However, 29.5% of patients developed asymptomatic grade 1 hypercalciuria without renal function changes or clinical symptoms, prompting discontinuation of VD supplementation in 10 out of 13 patients receiving adjuvant CT [7]. These findings suggest that while VD supplementation can be beneficial, monitoring for potential side effects such as hypercalciuria is essential.

Together, these studies illustrate the multifaceted role of VD in cancer treatment, ranging from its influence on bone density and side effect mitigation to its safety and tolerability in various oncological contexts. Transitioning between these diverse findings highlights the complexity of VD’s impact and the need for further research to optimize its use in oncology.

## 4. Discussion

To the best of our knowledge, this is the first scoping review aimed at evaluating the role of vitamin D (VD) supplementation in the side effects of adjuvant and neoadjuvant treatment of RT and CT in oncological patients.

While the studies included in this review present heterogeneity in methodologies, cancer diagnoses, and methods of assessing outcomes, they collectively underscore the potential role of VD in improving treatment tolerability and outcomes. This diversity highlights the need for standardized protocols to evaluate the effects of VD supplementation in cancer treatment, which could facilitate more robust comparisons in future research. In this paper, a review of the existing evidence is presented.

Yamada et al. concluded that patients undergoing preoperative CRT are at risk for impaired bone density, with VD supplementation potentially mitigating this risk and decreasing the incidence of distant metastases [37].

Similarly, Laviano et al. demonstrated that VD-containing TMN provided a favorable safety profile for non-small-cell lung cancer patients receiving CT, reducing adverse events like diarrhea, neutropenia, and nausea compared with isocaloric controls [38]. These findings align with the emerging understanding of VD as an adjuvant that may alleviate common side effects associated with cancer treatments.

Alkan et al.’s study concluded that the prevalence of VDD is high in cancer patients due to low exposure to sunlight, being in palliative or adjuvant settings, and a history of gastrointestinal surgery. These factors were classified as risk factors for VDD [6]. This underscores the multifactorial etiology of VDD in oncology, necessitating a tailored approach to prevention and management.

Jennaro et al. further established a connection between VDD and chemotherapy-induced peripheral neuropathy (PN) in breast cancer patients, suggesting that addressing VDD before treatment may reduce PN and improve therapeutic outcomes [19].

Chartron et al., in turn, concluded that VDD is very frequent in breast cancer patients and that this condition influences prognosis since taxanes, corticosteroids, behavioral changes (such as lack of sunlight exposure and poor dietary intake), and treatment-associated gastrointestinal side effects play a role in this initial decrease in VD levels [7].

They also concluded that VD supplementation increased the percentage of normalization of serum 25-OH VD concentration in breast cancer patients undergoing adjuvant CT and that this supplementation appears safe and effective [7].

A narrative review on patients undergoing hematopoietic stem cell transplantation (HSCT) further broadens the understanding of VD’s effects on severe diseases. The review highlighted that low VD levels are common in HSCT patients, both pre- and post-transplant, and are associated with an increased risk of developing graft-versus-host disease (GVHD), particularly in its chronic form. However, the relationship with acute GVHD was inconsistent. Supplementation protocols varied significantly, but VD plays a relevant immunomodulatory role, influencing GVHD pathogenesis. These findings suggest that VD deficiency in HSCT patients may compromise transplantation outcomes, emphasizing the importance of standardizing supplementation in this context [39].

Beyond oncology, other studies have explored the role of VD in chronic diseases, providing valuable insights that reinforce the findings of this review. Research into COVID-19 suggests that VD plays a protective role, with studies reporting a correlation between adequate VD levels and reduced disease severity, mortality, and hospitalization times [40,41]. Although these findings stem from a different context, they highlight VD’s potential in modulating immune responses and mitigating disease-related complications [41].

In other chronic conditions, VD’s anticancer mechanisms have been linked to its interaction with the vitamin D receptor (VDR) and its modulation of gene expression. These mechanisms, which include anti-inflammatory, antioxidant, and DNA repair activities, are also influenced by dysregulation in cancer [42]. Specifically, studies in prostate cancer reveal that VD analogs and combinations with other therapies may enhance anticancer effects, such as cell cycle arrest and apoptosis, although challenges like resistance to VD-based therapies remain [43].

Emerging therapeutic strategies also explore VD’s role in enhancing photodynamic therapy (PDT) efficacy. In skin cancer, VD has been shown to increase protoporphyrin IX (PpIX) accumulation, boost immune responses, and improve treatment efficacy in combination with PDT [44,45]. Such findings suggest that VD’s benefits extend beyond systemic supplementation to novel combinatory approaches [44].

The VITAL trial, a landmark study, showed no reduction in cancer incidence or mortality with VD supplementation in healthy adults [46]. However, the trial’s focus on primary prevention rather than treatment or supplementation in deficient individuals highlights the need for further context-specific investigations [47].

Finally, evidence from Japanese breast cancer patients treated with neoadjuvant chemotherapy suggests that serum VD levels may influence prognosis, emphasizing the importance of monitoring and optimizing VD intake in oncological care [46].

In conclusion, this scoping review, supported by broader literature, underscores VD’s potential role in mitigating cancer treatment side effects and improving patient outcomes. However, significant gaps remain, particularly regarding standardization in methodologies and the long-term impact of supplementation. Future research should prioritize these areas, exploring the broader applicability of VD-based interventions across cancer types and chronic diseases to fully realize their therapeutic potential.

### 4.1. Strengths and Limitations

This scoping review has several strengths and limitations. It involved an extensive research strategy and detailed screening to capture the breadth and depth of the literature in this field. To date, it is the first scoping review carried out on this topic, making it important for future clinical practice. An interdisciplinary team of nutritionists and researchers offering unique and complementary knowledge and perspectives completed it.

Some factors in the methodology of this review may have contributed to the limitations of its results, namely the definition of very strict inclusion and exclusion criteria, including the time limit of the last five years, which is why only five publications were selected. In addition, even though this review used three databases, searching other databases, such as the Web of Science, Mediclatina, and Embsase, may have yielded other published papers relevant to the purposes of this scoping review. However, it is important to note that access to Embase was not possible due to the lack of a subscription by the university. This review was also limited to papers in Portuguese, English, Spanish, and French. Other relevant papers on the subject may have been written in other languages and additional databases.

Moreover, the majority of the studies included a small sample without addressing whether the sample size was sufficient to detect an association, thus jeopardizing the generalizability of findings. Although the review is comprehensive, the limited sample sizes in the included studies restrict the generalization of results. Future research should prioritize larger cohort studies to address this limitation.

Thus, further studies can and should be conducted with the aim of better understanding the role of VD in the side effects of this treatment. Another limitation is that a critical analysis and bias assessment of the articles were not carried out, which may impact the interpretation and reliability of the included findings.

### 4.2. Future Directions

Future research should include longitudinal studies to evaluate the ongoing benefits or risks of VD supplementation in oncological patients. It is also essential to investigate the biological mechanisms by which VD modulates the side effects of chemotherapy and radiotherapy. Furthermore, multicenter randomized clinical trials should be designed to validate the efficacy and safety of VD supplementation in different cancer types and stages. In addition to clinical outcomes, these studies should incorporate holistic measures of impact, including assessments of quality of life and systemic health biomarkers, to better capture the full effects of VD supplementation.

Moreover, it is important to address existing research into VDD in the context of COVID-19.

## 5. Conclusions

In conclusion, the results of this scoping review suggest that VD supplementation may have a beneficial effect on the adverse effects associated with neoadjuvant treatment during RT or CT of oncological patients, both in the prevention and treatment of other health problems associated with treatment, such as osteoporosis, and in the mitigation of effects (nausea, vomiting, fatigue) resulting from aggressive CT and RT treatments.

However, few studies were found that directly link VD supplementation with the adverse effects of neoadjuvant treatment. Therefore, further studies are needed on this theme, particularly with more diversified samples regarding the age range and incidence of adverse effects associated with adjuvant and neoadjuvant RT and CT treatment of the oncological patient.

## Figures and Tables

**Figure 1 nutrients-17-00634-f001:**
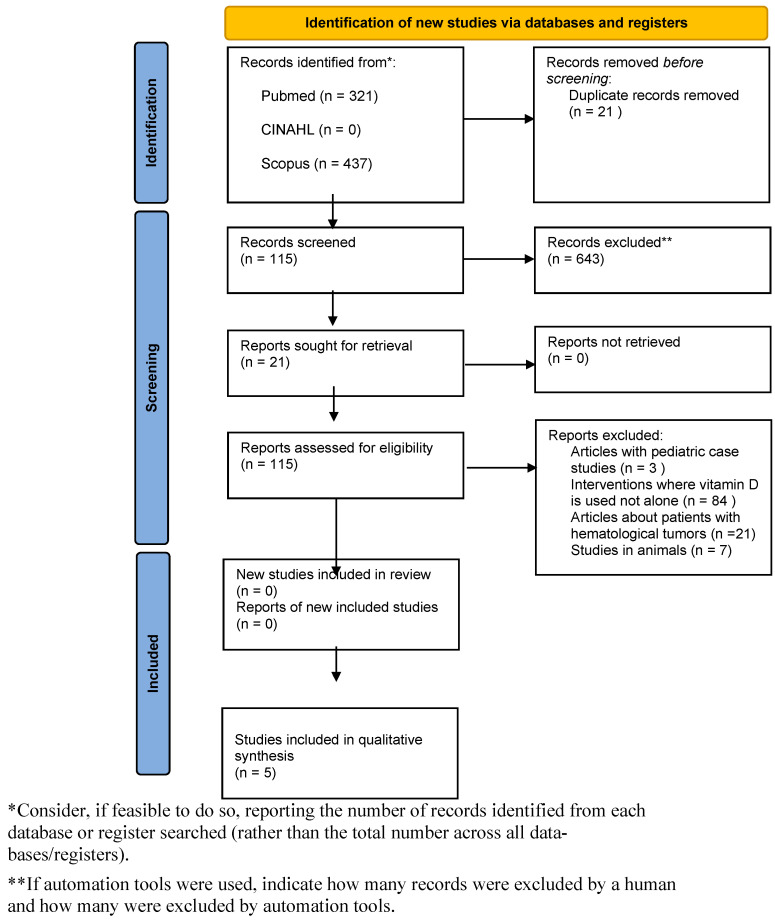
PRISMA-ScR 2020 study selection flow diagram.

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
