# Peer review of "A Systematic Review of Vitamin D Supplementation in Oncology: Chance of Science or Effectiveness?"

_nutrients, 2025, doi:10.3390/nu17040634_

Round 1
Reviewer 1 Report
Comments and Suggestions for Authors
Interesting and important article recommended for publication after considering my comments:
Article on Vitamin D Deficiency in Cornea
Review and Recommendations
This manuscript provides a comprehensive and important review of the role of vitamin D (VD) supplementation in reducing the side effects of adjuvant and neoadjuvant therapies in oncology patients. The article is well organized and addresses an understudied but clinically relevant topic. The following are key observations and recommendations:
Strengths:
Relevance of the topic: The focus on VD in oncology therapies is consistent with the growing need to effectively manage the side effects of therapy and improve patient outcomes.
Methodological rigor: The authors carefully implemented the PRISMA guidelines, ensuring a systematic approach to literature selection and review.
Interdisciplinary collaboration: The study benefits from diverse expertise, enhancing the depth of discussion and findings.
Areas for improvement:
It is important to note limited sample size in included studies: While the review is comprehensive, the small sample sizes in included studies limit generalizability. Future research should prioritize larger cohort studies.
Note diversity in study populations: The review primarily reflects data from specific demographics. Expanding to include studies with diverse populations would improve applicability to different patient groups.
Note language and database limitations: The limitation to four languages ​​and three databases likely excluded some relevant studies. Future reviews could incorporate additional databases (e.g., Web of Science, Embase) and obtain studies in other languages.
Note heterogeneity in study design: Significant methodological differences between the studies reviewed make comparison challenging. Standardized protocols for assessing the impact of VD on treatment outcomes should be encouraged.
Indicate recommendations for future research:
Longitudinal studies: Conduct long-term studies to assess ongoing benefits or risks of VD supplementation in oncology patients.
Mechanistic studies: Investigate the biological mechanisms by which VD modulates side effects of chemotherapy and radiotherapy.
Clinical trials: Design multicenter randomized controlled trials to validate the efficacy and safety of VD supplementation across different cancer types and stages.
Indicate holistic outcome measures: Include quality of life assessments and systemic health biomarkers alongside clinical endpoints to capture the full impact of VD supplementation.
It is important to address articles on vitamin D deficiency in COVID-19 such as:
The role of vitamin D deficiency on COVID-19: a systematic review and meta-analysis of observational studies
Associations of the COVID-19 burden and various comorbidities of different ethnic groups in Israel: a cross-sectional study
Vitamin D Deficiency in COVID-19 Patients and Role of Calcifediol Supplementation
Editorial suggestions:
It is important to have a detailed methodological explanation: Ensure that the methods section comprehensively explains the exclusion criteria, especially for complex cases such as hematological tumors or interventions that combine VD with other factors.
Availability of supplementary data: Encourage authors to provide detailed supplementary materials, such as search strategies or raw data tables, for increased transparency.
Clear future directions: Suggest including a dedicated section summarizing insights and actionable hypotheses generated by this review.
Author Response
Dear Editor,
Thank you for the opportunity to resubmit a revised version of this manuscript entitled “A Systematic Review of Vitamin D supplementation in oncology: chance of science or effectiveness?”
We have carefully reviewed the comments and have made every attempt to fully address them accordingly. Our responses are given in a point-by-point manner below.
We now believe these insightful revisions have resulted in a significantly improved manuscript and hope the revised version is now suitable for publication in the Special issue: Diet, Nutrition, Supplements and Integrative Oncology in Cancer Care of Nutrients. We look forward to hearing from you in due course.
Yours Sincerely,
Summary
We thank you for the time and effort that you dedicated on our manuscript, and we are grateful for the insightful comments to improve our paper.
Please see below, in blue, for a point-by-point response to the comments and concerns.
Review and Recommendations
This manuscript provides a comprehensive and important review of the role of vitamin D (VD) supplementation in reducing the side effects of adjuvant and neoadjuvant therapies in oncology patients. The article is well organized and addresses an understudied but clinically relevant topic. The following are key observations and recommendations:
Strengths:
Relevance of the topic: The focus on VD in oncology therapies is consistent with the growing need to effectively manage the side effects of therapy and improve patient outcomes.
Methodological rigor: The authors carefully implemented the PRISMA guidelines, ensuring a systematic approach to literature selection and review.
Interdisciplinary collaboration: The study benefits from diverse expertise, enhancing the depth of discussion and findings.
Author’s response - We thank the Reviewer for the positive feedback and for taking the time to review our work.
Areas for improvement
Comment 1 - It is important to note limited sample size in included studies: While the review is comprehensive, the small sample sizes in included studies limit generalizability.
Future research should prioritize larger cohort studies.
Author’s response - We appreciate your comment and understand that the sample size is a limitation of the study. Therefore, we have added more information regarding this topic on page 8, Section 4.1 - Strengths and limitations, 3rd paragraph, lines 335 to 339 ("Moreover ... to address this limitation").
Comment 2 - Note diversity in study populations: The review primarily reflects data from specific demographics. Expanding to include studies with diverse populations would improve applicability to different patient groups.
Author’s response – We agree with this comment and recognize the heterogeneity of the studies presented such as diversity in study populations. This information to clarify this topic can be found on page 6, Section 4. – Discussion, 2nd paragraph, lines 250 to 255 (“While the studies … existing evidence is presented”).
Comment 3 - Note language and database limitations: The limitation to four languages and three databases likely excluded some relevant studies. Future reviews could incorporate additional databases (e.g., Web of Science, Embase) and obtain studies in other languages.
Author’s response - We appreciate your suggestion, as we believe it enhances the interpretation of our findings and provides valuable considerations for future research in this field. We acknowledge that limiting our search to specific databases and four languages may have restricted the scope of our review and excluded potentially relevant studies. We have addressed this limitation on page 8, Section 4.1 – Strengths and Limitations, 2nd paragraph, lines 325 to 334 (“Some factors in the methodology … languages and additional databases”).
Comment 4 - Note heterogeneity in study design: Significant methodological differences between the studies reviewed make comparison challenging. Standardized protocols for assessing the impact of VD on treatment outcomes should be encouraged.
Author’s response - We agree with this comment and recognize the heterogeneity of the studies presented, particularly in their methodology. Information to clarify this topic can be found on page 6, Section 4. – Discussion, 2nd paragraph, lines 250 to 255 (“While the studies … existing evidence is presented”).
We also recognize the bias of the studies that were not considered, potentially impacting the results. This information can be seen on page 8, Section 4.1 – Strengths and Limitations, 4th paragraph, lines 340 to 343 (“Thus, further studies … reliability of the included findings”)
Indicate recommendations for future research:
Comment 5 - Longitudinal studies: Conduct long-term studies to assess ongoing benefits or risks of VD supplementation in oncology patients.
Mechanistic studies: Investigate the biological mechanisms by which VD modulates side effects of chemotherapy and radiotherapy.
Clinical trials: Design multicenter randomized controlled trials to validate the efficacy and safety of VD supplementation across different cancer types and stages.
Indicate holistic outcome measures: Include quality of life assessments and systemic health biomarkers alongside clinical endpoints to capture the full impact of VD supplementation.
It is important to address articles on vitamin D deficiency in COVID-19 such as:
The role of vitamin D deficiency on COVID-19: a systematic review and meta-analysis of observational studies
Associations of the COVID-19 burden and various comorbidities of different ethnic groups in Israel: a cross-sectional study
Vitamin D Deficiency in COVID-19 Patients and Role of Calcifediol Supplementation
Author’s response - We appreciate your recommendations. So, we have now added the section "Future Directions", which addresses these topics, on page 8, section 4.2. - Future directions, 1st and 2nd paragraphs, lines 345 to 354 (“Future research should … VDD in the context of COVID-19.”).
In order to include the relationship between Vitamin D and covid-19, section 4. Discussion was improved and some of these articles were mentioned, as can be seen on page 7, section 4. Discussion, 10th paragraph, lines 288 to 293 (“Beyond oncology … disease-related complications”).
Editorial suggestions:
Comment 6 - It is important to have a detailed methodological explanation: Ensure that the methods section comprehensively explains the exclusion criteria, especially for complex cases such as hematological tumors or interventions that combine VD with other factors.
Author’s response - We agree with your comment and have provided further clarification on the selection criteria to enhance the understanding of this topic. We have explained the restriction to studies published in the last five years (page 4, section 2.2. - Selection criteria, 3rd paragraph, lines 159 to 163 – “Focusing on … important in a systematic review”), the relevance of the topic (page 4, section 2.2. - Selection criteria, 4th paragraph, lines 164 to 167 – “This approach is especially … applications, including oncology”) the exclusion of articles involving patients with hematological tumors (page 4, section 2.2. - Selection criteria, 5th paragraph, lines 168 to 172 – “As exclusion criteria … analysis between studies difficult”), and the exclusion of studies where Vitamin D intervention was not used in isolation (page 4, section 2.2. - Selection criteria, 5th paragraph, lines 172 to 176 – “In addition… specific role of vitamin D”) .
Comment 7 - Availability of supplementary data: Encourage authors to provide detailed supplementary materials, such as search strategies or raw data tables, for increased transparency.
Author’s response - We acknowledge the importance of providing supplementary materials, such as search strategies or raw data tables, to enhance the transparency of our research. Unfortunately, we no longer have these supplementary materials available, and we sincerely apologize for this limitation. We recognize that such information would contribute to the reproducibility and clarity of our study and will ensure to incorporate it in future research endeavors.
Comment 8 - Clear future directions: Suggest including a dedicated section summarizing insights and actionable hypotheses generated by this review.
Author’s response - In response to your insightful comment, we have added a dedicated “Future Directions” section to the manuscript, summarizing key insights and actionable hypotheses generated by this review (page 8, section 4.2. - Future directions, 1st and 2nd paragraphs, lines 344 to 354 (“Future research should … VDD in the context of COVID-19.”).
Additionally, in the Conclusion section, we have emphasized the need for further studies on this topic, particularly those incorporating more diverse study samples (page 8, section 5.- Conclusions, 2nd paragraph, lines 361 to 365 – “However, few studies … treatment of the oncological patient”) . We believe these additions strengthen the discussion and provide a clearer perspective for future research in this field.
Reviewer 2 Report
Comments and Suggestions for Authors
In the present time, people in highly developed countries put their special attention to their helth condition. Moreover, the mental condition and external look are also in the mein point of interest. Therefore, they are looking for the magic pills which resolve their problem. The TV and internet supplement diet promotion mention only of their positive influence/aspects on health. The above has been positively presented to the opinion in social media. There is no doubt that this media coverage promotes only positive influence. Therefore, scientists are obligated to provide community education in a propre, objective manner. Due to that, with great pleasure, I read and analysed the sent article entitled: Vitamin D supplementation in oncology: chance of science or effectiveness? A scoping review. In this manuscript, authors discussed the cons and pros of Vit D consumption, especially in the case of of their derivatives. Moreover, the negative impact on oncologic patients has also been discussed.
The article is well written. The literature selection has been correctly focused and described.
I would be grateful if the authors could declare that artificial intelligence and ChatGPT have not been used.
Author Response
Dear Editor,
Thank you for the opportunity to resubmit a revised version of this manuscript entitled “A Systematic Review of Vitamin D supplementation in oncology: chance of science or effectiveness?”
We have carefully reviewed the comments and have made every attempt to fully address them accordingly. Our responses are given in a point-by-point manner below.
We now believe these insightful revisions have resulted in a significantly improved manuscript and hope the revised version is now suitable for publication in the Special issue: Diet, Nutrition, Supplements and Integrative Oncology in Cancer Care of Nutrients. We look forward to hearing from you in due course.
Yours Sincerely,
Summary
We thank you for the time and effort that you dedicated on our manuscript, and we are grateful for the insightful comments to improve our paper.
Please see below, in blue, for a point-by-point response to the comments and concerns.
Review and Recommendations
In the present time, people in highly developed countries put their special attention to their helth condition. Moreover, the mental condition and external look are also in the mein point of interest. Therefore, they are looking for the magic pills which resolve their problem. The TV and internet supplement diet promotion mention only of their positive influence/aspects on health. The above has been positively presented to the opinion in social media. There is no doubt that this media coverage promotes only positive influence. Therefore, scientists are obligated to provide community education in a propre, objective manner. Due to that, with great pleasure, I read and analysed the sent article entitled: Vitamin D supplementation in oncology: chance of science or effectiveness? A scoping review. In this manuscript, authors discussed the cons and pros of Vit D consumption, especially in the case of of their derivatives. Moreover, the negative impact on oncologic patients has also been discussed.
The article is well written. The literature selection has been correctly focused and described.
Author’s response - We appreciate your thoughtful and detailed feedback on our manuscript. We are grateful for your positive remarks regarding the clarity of our writing and the appropriate selection of literature.
Our goal was to present a comprehensive discussion on the benefits and potential risks of Vitamin D supplementation, particularly in oncology, to ensure a nuanced understanding of its implications.
Thank you once again for your insightful comments and for taking the time to review our work. Your feedback is greatly appreciated.
Comments 1 - I would be grateful if the authors could declare that artificial intelligence and ChatGPT have not been used.
Author’s response - We appreciate your comment, so we added this information on page 9, 1st paragraph, lines 371 and 372 (“All authors have … ChatGPT have not been used”).
Reviewer 3 Report
Comments and Suggestions for Authors
Dear Authors,
I believe that your manuscript addresses a highly impactful topic and aligns well with the objectives of the journal Nutrients. Below are my comments and suggestions:
ABSTRACT: Line 36: Instead of mentioning PRISMA for systematic reviews, please specify PRISMA ScR, which is tailored for scoping reviews.
INTRODUCTION: I do not have any comments, as the introduction is well-written and supported by adequate references.
METHODS:
- Table 1 should be included as a supplementary file, as it makes the main text too lengthy.
- The research question's PCC (Population, Concept, and Context) needs to be defined, as it is only mentioned.
- The rationale for conducting the search within the last 5 years should be clarified.
- The protocol for your scoping review should be registered. To ensure methodological quality, I recommend registering it on the Open Science Framework and including the DOI of the registration in the methods section.
RESULTS:
- Table 2 should be expanded by including the country, separating the aim from the intervention, and providing the results in a more standardized format.
- While the results are presented with a great deal of information, which is commendable, the text should be made more linear and better connected between paragraphs with transition phrases to improve readability.
DISCUSSION AND CONCLUSIONS:
- The discussion should be expanded. Specifically, your results should be further discussed and supported by other international studies, potentially comparing them with other chronic diseases. For instance, I suggest comparing with hematological diseases by referencing a recent study that explored the role of vitamin D in hematologic patients undergoing stem cell transplantation (DOI: 10.3390/nu16172976).
- Please review the final references as some are incorrectly reported
Author Response
Dear Editor,
Thank you for the opportunity to resubmit a revised version of this manuscript entitled “A Systematic Review of Vitamin D supplementation in oncology: chance of science or effectiveness?”
We have carefully reviewed the comments and have made every attempt to fully address them accordingly. Our responses are given in a point-by-point manner below.
We now believe these insightful revisions have resulted in a significantly improved manuscript and hope the revised version is now suitable for publication in the Special issue: Diet, Nutrition, Supplements and Integrative Oncology in Cancer Care of Nutrients. We look forward to hearing from you in due course.
Yours Sincerely,
Summary
We thank you for the time and effort that you dedicated on our manuscript, and we are grateful for the insightful comments to improve our paper.
Please see below, in blue, for a point-by-point response to the comments and concerns.
Review and Recommendations
Dear Authors,
I believe that your manuscript addresses a highly impactful topic and aligns well with the objectives of the journal Nutrients.
Author’s response - we thank so much the reviewer for the positive feedback.
Below are my comments and suggestions:
Comment 1 - ABSTRACT: Line 36: Instead of mentioning PRISMA for systematic reviews, please specify PRISMA ScR, which is tailored for scoping reviews.
Author’s response - Thank you for your suggestion. We have made the necessary change and specified PRISMA-ScR instead of PRISMA in the Abstract (page 1, Line 37).
Comment 2 - INTRODUCTION: I do not have any comments, as the introduction is well-written and supported by adequate references.
Author’s response - We appreciate the positive feedback on the introduction.
METHODS:
Comment 3 - Table 1 should be included as a supplementary file, as it makes the main text too lengthy.
Author’s response - Thank you for your helpful suggestion. As requested, we have moved Table 1 to the supplementary materials and included it as Appendix A in the manuscript. It can now be found on pages 9, 10, and 11, under the section Appendix A – Search terms used on PubMed, CINAHL, and Scopus databases.
We appreciate your feedback, which has helped improve the readability and organization of our manuscript.
Comment 4 - The research question's PCC (Population, Concept, and Context) needs to be defined, as it is only mentioned.
Author’s response - We agree and have added this information on page 3, section 2.1. Search strategy, 2nd paragraph, lines 132 to 135 (“In this review … radiotherapy and chemotherapy”).
Comment 5 - The rationale for conducting the search within the last 5 years should be clarified.
Author’s response - We appreciate your comment and clarify this topic on the page 4, section 2.2. Selection criteria, 3rd paragraph, lines 159 to 163 (“Focusing on recent studies … important in a systematic review”)
Comment 6 - The protocol for your scoping review should be registered. To ensure methodological quality, I recommend registering it on the Open Science Framework and including the DOI of the registration in the methods section.
Author’s response - Thank you for your valuable suggestion. We fully agree with your recommendation and will proceed with registering our scoping review on the Open Science Framework. Once completed, we will include the DOI of the registration in the Methods section. We appreciate your guidance in ensuring the methodological quality of our study.
RESULTS:
Comment 7 - Table 2 should be expanded by including the country, separating the aim from the intervention, and providing the results in a more standardized format.
Author’s response - Thank you for your insightful suggestion. We have revised Table 2 according to your recommendations by including the country, separating the aim from the intervention, and presenting the results in a more standardized format.
To improve readability, we have also moved Table 2 to the supplementary materials as Appendix B. It can now be found on pages 12 and 13 under the section Appendix B – “Summary of the selected studies”.
Comment 8 - While the results are presented with a great deal of information, which is commendable, the text should be made more linear and better connected between paragraphs with transition phrases to improve readability.
Author’s response - Thank you for your constructive feedback. We have revised the Results section to improve its readability and flow. The text has been restructured to make it more linear, with better connections between paragraphs using transition phrases. We believe these changes enhance the clarity and coherence of the section, as you suggested. You can see the changes on the page 4,5 and 6, section 3. Results, from the 2nd to the 7th paragraph, lines 199 to 245 (“Yamada et al. observed …to optimize its use in oncology”).
DISCUSSION AND CONCLUSIONS:
The discussion should be expanded. Specifically, your results should be further discussed and supported by other international studies, potentially comparing them with other chronic diseases. For instance, I suggest comparing with hematological diseases by referencing a recent study that explored the role of vitamin D in hematologic patients undergoing stem cell transplantation (DOI: 10.3390/nu16172976).
Author’s response - Thank you for your valuable suggestions. We have expanded the Discussion section as recommended, incorporating additional insights and comparisons, not only with other chronic diseases but also with other relevant areas that we believe contribute to the understanding of our findings. Page 7, section 4. Discussion, 10th to 15th paragraphs, lines 279 to 318 (“A narrative review … their therapeutic potential”).
Specifically, we have included a comparison with hematological diseases and referenced the study you suggested (DOI: 10.3390/nu16172976) on the role of vitamin D in hematologic patients undergoing stem cell transplantation. We hope these additions enrich the discussion and provide a broader context for our research. Page 7, section 4. Discussion, 9th paragraph, lines 279 to 287 (“A narrative review … supplementation in this context”).
Please review the final references as some are incorrectly reported
Author’s response - We appreciate your comment and we correct all references. Pages 14 to 17, references section.
Round 2
Reviewer 3 Report
Comments and Suggestions for Authors
The manuscript may be accepted for publication